# Genome-Wide Identification, Conservation, and Expression Pattern Analyses of the BBR-BPC Gene Family Under Abiotic Stress in *Brassica napus* L.

**DOI:** 10.3390/genes16010036

**Published:** 2024-12-29

**Authors:** Long Wang, Wei Chen, Zhi Zhao, Huaxin Li, Damei Pei, Zhen Huang, Hongyan Wang, Lu Xiao

**Affiliations:** 1Academy of Agricultural and Forestry Sciences, Qinghai University, Xining 810016, China; wanglong5898@163.com (L.W.); 18324703083@163.com (W.C.); zhaozhi918@sohu.com (Z.Z.); lhuaxin2000@163.com (H.L.); pdm2022@163.com (D.P.); 2Laboratory for Research and Utilization of Qinghai Tibet Plateau Germplasm Resources, Xining 810016, China; 3Key Laboratory of Spring Rapeseed Genetic Improvement of Qinghai Province, Xining 810016, China; 4Qinghai Spring Rape Engineering Research Center, Xining 810016, China; 5State Key Laboratory of Crop Stress Biology for Arid Areas, College of Agronomy, Northwest A&F University, Xianyang 712100, China; huang_zhen.8@163.com; 6Laboratory of Plant Epigenetics and Evolution, School of Life Science, Liaoning University, Shenyang 110036, China

**Keywords:** BBR-BPC, expression pattern, evolution, conservation, *Brassica napus* L.

## Abstract

Background: The BBR-BPC gene family is a relatively conservative group of transcription factors, playing a key role in plant morphogenesis, organ development, and responses to abiotic stress. *Brassica napus* L. (*B. napus*), commonly known as oilseed rape, is an allopolyploid plant formed by the hybridization and polyploidization of *Brassica rapa* L. (*B. rapa*) and *Brassica oleracea* L. (*B. oleracea*), and is one of the most important oil crops. However, little is known about the characteristics, conservation, and expression patterns of this gene family in *B. napus*, especially under abiotic stress. Methods: To explore the characteristics and potential biological roles of the BBR-BPC gene family members in *B. napus*, we conducted identification based on bioinformatics and comparative genomics methods. We further analyzed the expression patterns through RNA-seq and qRT-PCR. Results: We identified 25 BBR-BPC members, which were classified into three subfamilies based on phylogenetic analysis, and found them to be highly conserved in both monocots and dicots. The conserved motifs revealed that most members contained Motif 1, Motif 2, Motif 4, and Motif 8. After whole-genome duplication (WGD), collinearity analysis showed that BBR-BPC genes underwent significant purifying selection. The promoters of most BBR-BPC genes contained cis-acting elements related to light response, hormone induction, and stress response. RNA-seq and qRT-PCR further indicated that *BnBBR-BPC7*, *BnBBR-BPC15*, *BnBBR-BPC20*, and *BnBBR-BPC25* might be key members of this family. Conclusions: This study provides a theoretical foundation for understanding the potential biological functions and roles of the BBR-BPC gene family, laying the groundwork for resistance breeding in *B. napus*.

## 1. Introduction

Polyploidization is a common phenomenon in plant diversification and evolution, which is of great importance for the speciation of angiosperms [1,2]. Compared to normal diploid plants, polyploid plants exhibit significant yield advantages and superior environmental adaptability, and have rich genetic diversity [3,4]. Due to their experience of chromosome doubling and recombination, this also provides more possibilities for plant evolution and the selection of resistant varieties [5]. Previous studies have shown that one class of proteins capable of binding to GAGA repeat sequences is called GBP proteins. Among them, BBR proteins also possess GAGA-binding properties and share similar conserved domains and functions with GBP proteins, classifying them as a type of transcription factor [6,7]. These proteins are also referred to as BPC proteins due to the presence of multiple conserved cysteine residues at their C-termini [8]. As BPC proteins have been explored in-depth, they are commonly referred to as BBR/BPC proteins. BBR-BPC (Barley B recombinant/Basic Penta cysteine) represents a unique class of transcription factors in plants, playing critical roles in organogenesis, ovule development, cell division and differentiation (senescence), and plant hormone responses [9,10,11]. Given their functional diversity and conservation in the life cycle of plants, these transcription factors have been extensively investigated in various plant species, including *Arabidopsis thaliana* L. [12], *Oryza sativa* L. [13], *Cucumis sativus* L. [14], *Prunus persica* L. [15], and *Camellia japonica* L. [16].

Currently, *BBR-BPC* genes have been identified in many plants, showing a degree of functional redundancy. It has been reported that there are seven *BBR-BPCs* in *A. thaliana*, which are crucial for regulating ovule development. Further analysis has shown that these can control ovule development by modulating the expression of the MADS-box domain containing gene *STK* [12]. However, loss of *BBR-BPC* gene function in *A. thaliana* results in dwarfism, abnormal ovule development, and reduced lateral root numbers [12]. *AtBBR-BPCs* can also lead to the abnormal development of floral organs by downregulating the expression of the *STM* and *BP* genes [17]. In *C*. *sativus*, *BBR-BPC* genes are also involved in seed germination, and it is found that the expression of *CsBBR-BPCs* significantly increases under abiotic and hormone stress [14]. Moreover, the knockdown of the *AtBBR-BPC2* gene in *A. thaliana* under osmotic stress showed increased stress tolerance, while plants overexpressing this gene displayed sensitivity [18]. Additionally, *BBR-BPC* genes play a vital role in regulating leaf morphology. For instance, the overexpression of *BBR-BPC* genes in tobacco alters leaf morphology [7]. In a recent study, the BBR-BPC gene family (*BBR-BPC3*) was found to play a key role in regulating the flowering process, diurnal rhythm changes, and salt stress response in plants [10,19,20]. Additionally, *BBR-BPC3* in *A. thaliana*, which regulates diurnal rhythms, has antagonistic interactions with *BBR-BPC3* in *O*. *sativa*. This regulates the flowering process and plant height and thus reveals that the BBR-BPC gene family plays different roles throughout the plant lifecycle.

BBR-BPC is a more conservative gene family, and plays an important role in plant growth and development as well as stress response. *B. napus*, a member of the *Brassicaceae* family within the genus *Brassica*, is one of the major economic crops, formed through the natural hybridization and polyploidization of *B. rapa* and *B. oleracea* [21,22,23]. However, as an allopolyploid plant, studying its environmental adaptability and evolution is particularly important [24]. Therefore, in this study, the BBR-BPC gene family was identified and explored through bioinformatics and comparative genomics and the characteristics, conservation, and expression patterns of different members were analyzed. These findings lay the foundation for the important roles of the BBR-BPC gene family members in plant growth, development, and environmental adaptation, and provide references for further understanding the evolution of polyploid plants and the selection of resistant varieties.

## 2. Materials and Methods

### 2.1. Whole-Genome Identification of BBR-BPC Gene Family Members

*B. napus* is a typical polyploid plant and an important oilseed crop. To determine the characteristics of the BBR-BPC gene family in *B. napus*, genomic information was obtained from the BRAD database [25]. At the same time, BBR-BPC sequence information from *A. thaliana* was downloaded from the TAIR database [26]. Initially, the protein sequences of AtBBR-BPC from *A. thaliana* were used as a reference for homology comparison in *B. napus*, with the parameters for E-value of less than 1 × 10^−5^ and homology ≥ 65% [27,28]. Subsequently, the protein sequences of *B. napus* were screened using the Hidden Markov Model (PF06217) from the Pfam database [29]. Sequences identified by both methods were retained and redundant sequences were removed. HMMER, SMART, and the NCBI-CDD search were utilized to identify conserved domains within the BBR-BPC proteins to ensure the accuracy of the candidate sequences. Ultimately, sequences exhibiting characteristics of BBR-BPC proteins were preserved for further analysis [30,31,32].

### 2.2. Physicochemical Properties and Chromosomal Localization

To analyze the physicochemical properties and subcellular localization of the BBR-BPC gene family members, we utilized the Expasy-ProtParam tool and the WoLF PSORT tool for calculations and predictions [33,34]. Based on the genome annotation files for *B. napus*, MG2C (MapGene2 Chromosome) software (version 2.1) was used to construct the chromosome lengths and gene positions [35].

### 2.3. Conservation Analysis of the BBR-BPC Gene Family

To further explore the conservation and evolutionary distinctions of the BBR-BPC gene family, the *B. napus* genome was analyzed using genomic files and annotations. CD-search and MEME Suite 5.5.0 were applied as tools to identify conserved domains, motifs, and gene structures. The findings were visualized through the Gene Structure View feature in TBtools software (version 1.120).

### 2.4. Phylogenetic and Collinearity Analysis

To analyze the phylogenetic relationship of the BBR-BPC gene family and investigate the duplication events involving BBR-BPC members in *B. napus*. The BBR-BPC protein sequences in *B. napus*, along with previously reported sequences from *Zea mays* L., *A. thaliana*, *Sorghum bicolor* L., and *O. sativa*, were aligned using MEGA-X for multiple-sequence alignment. Subsequently, a phylogenetic analysis was conducted using the Neighbor-Joining (NJ) method [36]. Collinearity analysis within *B. napus* and between *B. napus* and *A. thaliana* was conducted using MCScanX. The results were visualized using Circos (v0.69) [37]. Synonymous (Ks), non-synonymous (Ka), and the Ka/Ks ratios for each pair of duplicated genes were calculated using the Ka/Ks Calculator 2.0, as shown in Appendix A [38].

### 2.5. Cis-Acting Elements and KEGG Enrichment

To explore the key functional elements and enriched pathways in the BBR-BPC gene family members, an R script was used to extract sequence information within a 2000 bp range upstream of the promoter regions for members of the BBR-BPC gene family in *B. napus*. The extracted promoter sequences were then analyzed using the PlantCARE database [39]. Annotation information for BnBBR-BPC protein sequences was obtained using the eggNOG tool. Subsequently, KEGG enrichment analysis was performed using the ClusterProfiler package in R (version 4.2.1) [40].

### 2.6. Interaction Prediction and Subcellular Localization

To explore potential members that may interact with the BBR-BPC gene family, using the STRING protein interaction database (https://cn.string-db.org/, accessed on 6 March 2024), the interactions among *B. napus* BBR-BPC homologous proteins were predicted based on their *Arabidopsis* counterparts, with a significance level of *p*-value = 1.0 × 10^−16^. For subcellular localization, the PCAMBIA2300-GFP vector was used to transform Agrobacterium. Before infection, *Nicotiana tabacum* L. leaves were exposed to normal light to open their stomata. The leaves were then infected and observed under a confocal microscope 2–3 days post-infection to examine the infected areas.

### 2.7. Expression Pattern Analysis

To explore the potential biological functions of the BBR-BPC gene family, their expression patterns under various conditions were analyzed using transcriptome data from *B. napus*, available in the BnIR database [41]. These included data from different tissues (bud, filament, petal, pollen, sepal, cotyledon, vegetative rosette, root, lower stem peel, middle stem peel, and upper stem peel) and various developmental stages of siliques and seeds (from 14 DAF to 60 DAF). Expression patterns were visualized based on the Transcripts Per Million (TPM) values for each gene, using the Heatmap package in R.

### 2.8. Stress Response Analysis (Salt, Drought, Hormones)

In the experiment, ZS11 seeds were sterilized and planted in a growth chamber simulating natural light cycles (12 h light/12 h dark). When the seedlings reached four weeks of age, a saline stress environment was simulated using a 200 mmol/L sodium chloride solution, drought stress was mimicked using a 10% concentration of PEG-6000 solution, and hormone stress was simulated using a 200 mmol/L methyl jasmonate solution. Leaf samples were then collected at 0, 3, 6, 12, and 24 h after stress treatment and immediately flash-frozen in liquid nitrogen for subsequent RNA extraction.

To explore changes in gene expression, total RNA was extracted from the leaf samples using an RNA extraction kit (Sangon Biotech, Shanghai, China), followed by reverse transcription of RNA into cDNA using the PrimeScript™ RT Reagent Kit (Takara, Kusatsu, Japan). Specific primers for the target genes were designed using the NCBI primer-blast tool, and *BnActin7* (GeneBank ID: GBEQ01027912.1) was selected as an internal reference gene to normalize variations in gene expression (Appendix A). The qPCR SYBR Green Master Mix (Takara, Japan) was used in conjunction with the LightCycler 480 real-time PCR system to detect the fluorescence signals of each sample. Each PCR reaction included three technical replicates, and the relative expression levels of the genes were calculated using the 2^−ΔΔCT^ method based on the Ct values at a specific fluorescence threshold. Mean (±SE) expression values were calculated from three independent biological replicates and three technical replicates (*, *p* < 0.05; **, *p* < 0.01; ***, *p* < 0.001). The raw data are compiled in Appendix A.

## 3. Results

### 3.1. Identification and Characteristic Analysis of the BBR-BPC Family Members in B. napus

Through homology comparison and the validation of conserved domains using tools such as Pfam and SMART, a total of 25 members of the BBR-BPC transcription factor family were identified in *B. napus*, each possessing characteristic conserved domains. These members were distributed across 16 chromosomes in *B. napus*. Based on their chromosomal locations, they were named *BnBBR-BPC1* through *BnBBR-BPC25*. Physicochemical analysis revealed considerable variation among members of the BBR-BPC family (Table 1). Specifically, the number of amino acids ranged from 134 to 570, molecular weights varied from 14,968.89 Da to 65,338.51 Da, and the isoelectric points ranged from 8.66 to 10.46. Most proteins were identified as unstable (88%) and hydrophilic (100%), indicating a predisposition for interaction within cellular environments. Moreover, subcellular localization predictions suggested that most *BnBBR-BPC* members were located in the nucleus, with a minority found in the cytoplasm, underscoring their potential critical roles in the nucleus. The above results indicate that members of the BBR-BPC family are basic proteins, most of which exhibit hydrophilic and unstable characteristics. However, they are more likely to function in the nucleus.

### 3.2. Chromosomal Localization of the BBR-BPC Gene Family

In accordance with the genome annotation information of *B. napus*, a chromosomal location analysis of the 25 identified *BnBBR-BPC* genes revealed that most members were concentrated on chromosome Cn08, specifically *BnBBR-BPC22*, *BnBBR-BPC23*, *BnBBR-BPC24*, and *BnBBR-BPC25*. Chromosomes An04, An06, An08, An09, Cn03, and Cn04 each had two *BnBBR-BPC* members distributed evenly, while the remaining chromosomes each had one *BnBBR-BPC* member (Figure 1). Additionally, a comparison of the family members’ locations in the A and C sub-genomes of *B. napus* showed a highly similar quantity and distribution pattern of *BnBBR-BPC* family members, with 12 members in the A sub-genome and 13 in the C sub-genome. This suggests that the family members may be conserved across their diploid ancestors (*B. rapa* and *B. oleracea*) and the allopolyploid *B. napus*.

### 3.3. Phylogenetic Analysis of the BBR-BPC Gene Family

To understand the evolution and phylogenetic relationships of the BBR-BPC gene family, a phylogenetic tree incorporating BBR-BPC protein sequences from species such as *B. napus*, *A. thaliana*, *Z. mays*, *S. bicolor*, and *O. sativa* was constructed (Figure 2). The results indicate that the BBR-BPC gene family can be divided into five subfamilies, in which Subfamilies I, IV, and V contained the most members, and Subfamilies II and III contained fewer members. In addition, each subfamily in *B. napus* includes representative *A. thaliana* sequences: Subfamily I includes AtBBR-BPC4, AtBBR-BPC5, and AtBBR-BPC6; Subfamily III includes AtBBR-BPC7; and Subfamily V includes AtBBR-BPC1, AtBBR-BPC2, and AtBBR-BPC3, consistent with previous research. In the phylogenetic relationships, Subfamily I comprises BnBBR-BPC1, BnBBR-BPC4, BnBBR-BPC5, BnBBR-BPC7, BnBBR-BPC8, BnBBR-BPC9, BnBBR-BPC11, BnBBR-BPC13, BnBBR-BPC16, BnBBR-BPC17, BnBBR-BPC18, BnBBR-BPC20, BnBBR-BPC21, BnBBR-BPC23, and BnBBR-BPC24. Subfamily III includes BnBBR-BPC3 and BnBBR-BPC15, while Subfamily V includes BnBBR-BPC2, BnBBR-BPC6, BnBBR-BPC10, BnBBR-BPC12, BnBBR-BPC14, BnBBR-BPC19, BnBBR-BPC22, and BnBBR-BPC25. Furthermore, monocots like *Z. mays*, *S. bicolor*, and *O. sativa* (Subfamilies II and IV) cluster together, while dicots such as *A. thaliana* and *B. napus* (Subfamilies I, III, and V) cluster together. It can be observed that the BnBBR-BPC members are divided into three subfamilies, with the members of Subfamily I being the most abundant, and those of Subfamily III the least. Furthermore, the BBR-BPC gene family may be conserved in both monocots and dicots.

### 3.4. Motif Composition, Conserved Structural Domains, and Gene Structures of the BBR-BPC Gene Family

To understand the conservation and evolutionary traits of the BBR-BPC gene family, their conserved domains, motifs, and gene structures were analyzed, as depicted in Figure 3. As shown in Figure 3a, all members of the BBR-BPC gene family contain the characteristic GAGA-binding conserved domain. Analysis of the conserved motifs indicates that members within the same subfamily share similar motif compositions, whereas different subfamilies exhibit distinct motifs (Figure 3b). Specifically, Subfamily I predominantly consists of seven motifs: Motif 1, Motif 2, Motif 3, Motif 4, Motif 5, Motif 7, and Motif 8, with Motif 3 and Motif 5 being unique to this subfamily. In contrast, the motif composition of subfamily III members is somewhat simpler, including only four key motifs: Motif 1, Motif 2, Motif 4, and Motif 8. Subfamily V primarily features six motifs: Motif 1, Motif 2, Motif 4, Motif 6, Motif 7, and Motif 8, with Motif 6 unique to this subfamily. These results suggest that members within the same subfamily often share similar motifs, indicating likely similar gene functions. However, unique motifs in different subfamilies, such as Motif 3 and Motif 5 in Subfamily I and Motif 6 in Subfamily V, may explain the diverse roles between subfamilies and provide clues to their functional diversity. Moreover, four key motifs (Motif 1, Motif 2, Motif 4, and Motif 8) are present across different subfamilies, suggesting higher conservation and a significant role in the growth and development of *B. napus*. Gene structure analysis (Figure 3c) shows that most members have 1–2 exons, consisting of one shorter and one longer exon. Notably, *BnBBR-BPC14* exhibits an insertion of an approximately 30 kb intronic structure between its first and second exons, potentially due to incomplete genome assembly or annotation errors. The above results indicate that members of the BBR-BPC gene family all contain the characteristic GAGA-binding conserved domain. Additionally, Motif 1, Motif 2, Motif 4, and Motif 8 may represent relatively conserved motif elements. However, most members have relatively simple structures, containing only 1–2 exons.

### 3.5. Collinearity Analysis of the BBR-BPC Gene Family

Previous studies have revealed that gene duplication serves as a core driving force in the expansion of gene families, profoundly influencing the evolution of plant genomes and greatly enriching the genetic diversity and adaptive capacity of plants [42]. Consequently, interspecies and intraspecies collinearity analyses of the BBR-BPC gene family were conducted. As shown in Figure 4, gene duplication is widespread throughout the *B. napus* genome and is unevenly distributed across 16 chromosomes. However, chromosomes ChrA05, ChrA10, and ChrC09 show no collinear genes. Analysis revealed 50 pairs of collinear genes within *B. napus*. To further understand the evolutionary relationships of the BBR-BPC gene family within the *Brassicaceae*, we analyzed the collinearity between *B. napus* and *A. thaliana BBR-BPC* genes (Figure 5). Collinear genes in *B. napus* are distributed across 13 chromosomes, while in *A. thaliana*, they are distributed across three chromosomes, with 21 pairs of collinear genes identified between *B. napus* and *A. thaliana*. This indicates a high degree of gene exchange and similarity in the evolutionary process between these species. The interspecies and intraspecies collinearity analyses suggest that most of the collinear gene pairs are segmental duplications, indicating that segmental duplication events may play an important role in the evolution of the BBR-BPC gene family. To fully understand the impact of the BBR-BPC family genes on the evolution of *B. napus*, we analyzed the extent of selective pressure they experienced (Appendix A) [43]. All duplicated gene pairs have Ka/Ks ratios less than 1 (ranging from 0.028 to 0.423), indicating that the *B. napus* BBR-BPC family genes have undergone strong purifying selection and diversified early (2.2–36.0 Mya). The above results suggest that the BBR-BPC family members in *B. napus* mainly originate from segmental duplications and have undergone strong purifying selection during the evolutionary process.

### 3.6. Analysis of the Cis-Acting Elements in the Promoter Regions of the BnBBR-BPC Genes

Cis-acting elements in promoters are crucial for gene transcription and the regulation of gene expression [27]. To analyze the cis-acting elements in the promoters of the BBR-BPC gene family in *B. napus*, the 2 kb sequence upstream of the transcription start site for each gene was extracted, identifying 39 types of cis-acting elements (Figure 6). There are significant variations in the types and quantities of cis-acting elements in the promoters of the BBR-BPC family genes. For example, cis-acting elements related to plant growth and development (GT1-motif, G-box, Box4, TCT-motif), stress responses (ARE), and plant hormone responses (CGTCA-motif, TGACG-motif, ABRE) were notably enriched. By comparing the cis-acting elements across different subfamilies, we found that members of Subfamily I were enriched with elements related to plant growth and development (CAT-box, AE-box, GT1-motif, G-box, and TCT-motif), especially *BnBBR-BPC9*, *BnBBR-BPC16*, and *BnBBR-BPC24*. Additionally, cis-acting elements related to metabolism (O_2_-site) were more prevalent in Subfamilies I and V but less so in Subfamily III. Furthermore, in terms of response to abiotic stress, *BnBBR-BPC12* from Subfamily V contained a richer array of cis-acting elements compared to other subfamilies. Notably, TCT-motif, ARE, CGTCA-motif, TGACG-motif, and ABRE are widely present among different members of the BBR-BPC gene family. This indicates that the promoter regions of the BBR-BPC family members in *B. napus* are significantly enriched with cis-acting elements related to growth and development, hormone induction, and stress response. However, there are notable differences between subfamilies and between individual members. Furthermore, the widely distributed cis-acting elements may play an important role in regulating the basic life processes of *B. napus*.

### 3.7. KEGG Enrichment Analysis of BnBBR-BPC Genes

To uncover the biological functions of the BBR-BPC genes, we conducted an enrichment analysis of the 25 *BnBBR-BPC* genes in *B. napus*. The results indicated that *BnBBR-BPCs* were primarily enriched in areas such as molecular function, nucleic acid binding, cellular component, and DNA-binding transcription factor activity (Figure 7). The analysis of the genes along these pathways revealed that 20 genes from the *B. napus* BBR-BPC gene family were enriched, predominantly including members from Subfamily I (*BnBBR-BPC1*, *BnBBR-BPC5*, *BnBBR-BPC7*, *BnBBR-BPC8*, *BnBBR-BPC9*, *BnBBR-BPC13*, *BnBBR-BPC16*, *BnBBR-BPC17*, *BnBBR-BPC18*, *BnBBR-BPC20*, *BnBBR-BPC21*, *BnBBR-BPC23*, *BnBBR-BPC24*) as well as from Subfamily III (*BnBBR-BPC3*, *BnBBR-BPC15*) and from Subfamily V (*BnBBR-BPC2*, *BnBBR-BPC6*, *BnBBR-BPC12*, *BnBBR-BPC14*, *BnBBR-BPC19*) (Figure 7a). Further analysis of the significantly enriched pathways showed that the 10 genes enriched in the DNA-binding transcription factor activity pathway were also enriched in five other pathways (Figure 7b). These findings suggest that the BBR-BPC family may only require certain genes to exert biological functions, indicating the presence of some functional redundancy within this gene family.

### 3.8. Potential Interaction Prediction and Subcellular Localization Analysis of BnBBR-BPC Gene Family Members

Interaction predictions were performed for *A. thaliana* BBR-BPC homologous proteins using the STRING online database, selecting *B. napus* as the target species. As shown in Figure 8, there are 118 interaction relationships, 21 potential interacting proteins, with a local clustering coefficient of 0.937, and an enrichment *p*-value of 1.0 × 10^−16^. Among these twenty-one potential interacting proteins, the assortment includes six related to DNA recombination, repair, and replication, two MADS-box proteins, one catalase, one LOB domain protein, one RALF-like protein, one proton transmembrane transporter, one oxygen-evolving enhancer protein, one senescence/dehydration-associated protein, one glucosinolate-related protein, and one critical GATA transcription factor. Additionally, three proteins of unknown function appear in the interaction network. Previous studies have shown that BBR-BPC proteins can bind to the GA/TC motif, and there may be some antagonistic interaction between them. In the interaction network, we successfully identified a GATA transcription factor [8]. Furthermore, one study also found that they can regulate the expression of genes containing the MADS-box domain, thereby controlling ovule development [12], which indirectly suggests the key roles of two MADS-box proteins in the interaction network. In this network, BnBBR-BPC7 from the *B. napus* is particularly crucial, potentially interacting with 18 proteins. Such direct or indirect interactions could be significant for functional studies of the BnBBR-BPC gene family. The subcellular localization results (Figure 9) indicate that BnBBR-BPC7 protein shows fluorescence signaling in the nucleus, consistent with our predictions (Table 1). This suggests that BnBBR-BPC7 might be an important member that plays a key role in the nucleus.

### 3.9. Analysis of BnBBR-BPC Gene Expression Patterns

To explore the potential biological functions of different members of the BBR-BPC gene family, the tissue expression patterns of 25 BBR-BPC family members in various tissues of *B. napus* (ZS11) and during different developmental stages of seeds and siliques were analyzed (Figure 10). As shown in Figure 10a, the *B. napus BBR-BPC* genes are expressed across multiple tissues. Specifically, within Subfamily I, *BnBBR-BPC20* shows the highest expression in the stem peel (TPM > 40) and filament (TPM = 23); *BnBBR-BPC23* has the highest expression in the bud (TPM = 27); *BnBBR-BPC13* is most expressed in petal and sepal tissue (TPM = 27 and TPM = 25, respectively); and in Subfamily V, *BnBBR-BPC10* and *BnBBR-BPC25* show high expression levels in the root and pollen, respectively (TPM > 20). However, members of the BBR-BPC family generally exhibit low to no expression in cotyledon and vegetative rosette (TPM < 10). The developmental stages of siliques (Figure 10b) show that compared to members from other subfamilies (TPM < 10), members of Subfamily I (*BnBBR-BPC11*, *BnBBR-BPC23*) and Subfamily V (*BnBBR-BPC10*, *BnBBR-BPC22*) have relatively high expression levels (TPM > 10) and display an upregulation followed by a downregulation trend. Similarly, during the different developmental stages of seedlings (Figure 10c), members of Subfamily I (*BnBBR-BPC11*, *BnBBR-BPC23*) and Subfamily V (*BnBBR-BPC10, BnBBR-BPC12*, *BnBBR-BPC25*) also exhibit higher expression levels and a similar expression trend. These results suggest that different *BnBBR-BPC* genes may be involved in various growth and developmental processes in *B. napus*, with certain functional redundancies observed. However, it appears that members of Subfamilies I and V may play critical roles during the growth and development of the plant. This differential expression across tissues and developmental stages could indicate specialized functions tailored to specific physiological needs, reflecting the evolutionary diversification within *BnBBR-BPC*.

### 3.10. Real-Time PCR Analysis of BnBBR-BPC Family Members Under Salt Treatment

Considering the abundance of cis-acting elements related to stress and hormone responses in the BBR-BPC gene family, such as the Antioxidant Response Element (ARE) and hormone response elements (CGTCA-motif, TGACG-motif, and ABRE), their expression patterns under various treatments, including salt treatment, drought treatment, and hormone-induced treatment were analyzed. The results, as illustrated in Figure 11, show that under salt stress, compared to the control (0 h), members of Subfamily I generally exhibited an upregulation trend, reaching peak expression at 6 h or 24 h. Notably, *BnBBR-BPC20* showed a significant increase, with a relative expression level of 39.5. Members of Subfamily III (*BnBBR-BPC3* and *BnBBR-BPC15*) also showed an upregulation trend; however, the increase in *BnBBR-BPC15* was more pronounced compared to *BnBBR-BPC3*, suggesting that the latter might be more functionally significant in this subfamily. Most members of Subfamily V displayed an upregulation trend, with the highest relative expression levels observed at 6 h or 24 h, particularly for *BnBBR-BPC2*, *BnBBR-BPC10*, *BnBBR-BPC22*, and *BnBBR-BPC25*. The above results suggest that in response to salt stress, members of different subfamilies in B. napus exhibit distinct expression patterns. However, members in Subfamily V may show a stronger response to stress.

### 3.11. Real-Time PCR Analysis of BnBBR-BPC Family Members Under Drought Treatment

To further investigate the potential roles of *BnBBR-BPC* during abiotic stress processes, particularly under drought conditions, we conducted drought stress treatments and analyzed gene expression patterns using qRT-PCR technology (Figure 12). Compared to the control (0 h), members of Subfamily I predominantly exhibited an upregulation or an initial upregulation followed by a downregulation trend, with peak expressions occurring at 12 or 24 h of stress treatment. Notably, *BnBBR-BPC20* showed a particularly significant upregulation, with a relative expression level reaching 82.1, suggesting that this gene may be a key player within the BnBBR-BPC gene family. In contrast to the expression patterns observed in Subfamily I, members of Subfamilies III and V displayed similar expression profiles, also showing an initial upregulation followed by a downregulation trend. Except for *BnBBR-BPC14*, all other members in these subfamilies reached their peak expression at 12 h of stress treatment. Additionally, *BnBBR-BPC25* from Subfamily V reached a remarkably high relative expression level of 226 at 12 h under drought stress. The results indicate that *BnBBR-BPC20* and *BnBBR-BPC25* may play active roles in responding to drought stress.

### 3.12. Real-Time PCR Analysis of BnBBR-BPC Family Members Under Hormone Treatment

Given that the promoter regions of BBR-BPC gene family members are enriched with MeJA response elements, we analyzed the expression patterns of these genes under MeJA stress to identify potential key members of the BBR-BPC gene family. As shown in Figure 13, members of Subfamily I predominantly exhibited a trend of initial upregulation followed by downregulation, with their relative expression levels reaching a peak 12 h after hormone stress compared to the control group (0 h). Subfamily III includes two members, *BnBBR-BPC3* and *BnBBR-BPC15*, which displayed different expression patterns. *BnBBR-BPC3* showed a trend of initial downregulation followed by upregulation, while *BnBBR-BPC15* exhibited an initial upregulation followed by downregulation. Most members of Subfamily V initially showed an upregulation followed by a downregulation trend, but there were also cases of continuous downregulation (*BnBBR-BPC2*) and initial downregulation followed by upregulation (*BnBBR-BPC6* and *BnBBR-BPC19*). Notably, within this subfamily, *BnBBR-BPC25* reached its maximum relative expression level (24.3) at 6 h post-hormone treatment. The results indicate that members from different subfamilies exhibit varying expression patterns in response to hormone stress, particularly *BnBBR-BPC25*, which may play an active role.

## 4. Discussion

Polyploidization is a common event in species evolution and plays a significant role in plant evolution and speciation [21,22,23,44]. The BBR-BPC gene family is crucial in processes such as ovule development, flowering, diurnal rhythm changes, and responses to abiotic stress [19,20]. Although this gene family has been studied in *A. thaliana*, *O. sativa*, *C. sativus*, and *P. persica*, it has been explored less in the Brassicaceae family. Therefore, it is of great significance to characterize the BBR-BPC gene family and analyze its conservation in order to understand its role during the formation of *B. napus*.

Using homology comparison and conserved-domain identification methods, 25 *BBR-BPC* family members were ultimately identified in *B. napus*, with their numbers and distribution being highly similar across the A sub-genome (12) and C sub-genome (13). Further evolutionary analysis across different species revealed that *BBR-BPC* genes from monocots and dicots cluster into separate branches, suggesting a high degree of evolutionary conservation within the BBR-BPC gene family. Moreover, members within the same subfamily share similar conserved motifs, while notable differences exist between different subfamilies, providing clues to their functional divergences. The prevalence of polyploidy indicates that whole-genome duplication events may be significant in plant evolution. Interspecies and intraspecies collinearity analyses revealed that most genes were of the segmental duplication type, suggesting that segmental duplication events may play a crucial role in the evolution of the BBR-BPC gene family. Research has shown that redundant genes present in polyploid plants after whole-genome duplication (WGD) events can help them overcome environmental changes and stress damage, while also promoting evolutionary processes. However, the importance of WGD for evolution remains controversial. In *B. napus*, the primary form of gene duplication during the WGD event is segmental duplication. We believe that this WGD event may increase mutation probability and environmental adaptability due to functional redundancy or increased genetic variation. However, from a short-term or even recent perspective, the selective advantage of WGD is difficult to explain. The BBR-BPC gene family consists of transcription factors that can bind to the GA motif. KEGG enrichment analysis indicates that only 10 members of the BBR-BPC family are enriched in key DNA-binding pathways. However, these 10 members are also enriched in other pathways, further suggesting that the BBR-BPC gene family may require only a few members to perform their functions. Additionally, the analysis of the BBR-BPC family genes’ impact on the evolution of *B. napus* showed that their Ka/Ks values are all less than 1 (ranging from 0.042 to 0.676), indicating strong purifying selection throughout their evolution. The divergence times suggest that the BBR-BPC might be a relatively ancient gene family, dating back from 2.2 Mya to 36.0 Mya.

The types and quantities of cis-acting elements are essential for the transcriptional process and the regulatory control of gene expression [40]. The BBR-BPC gene family’s cis-acting elements were extensively analyzed, identifying 39 types of elements that were enriched to varying degrees in growth and development, stress response, and hormone response functions. Notably, Subfamily I is particularly enriched in elements related to plant growth, such as TCT-motif, CAT-box, and G-box. Compared to elements related to metabolism and stress response, Subfamily I also extensively accumulates hormone response-related cis-acting elements like CGTCA-motif, TGACG-motif, ABRE, and TCA-element. This enrichment suggests that Subfamily I may play a significant role in plant growth and hormone signal transduction. Transcriptomic data reveal that within Subfamily I, *BnBBR-BPC13* exhibits the highest expression in the petal and sepal (TPM = 27, TPM = 25), and *BnBBR-BPC20* shows the highest expression in the stem peel (TPM > 40). Similarly, *BnBBR-BPC20* and *BnBBR-BPC23* have the highest expression in the filament and bud (TPM > 20), respectively. This suggests that the abundance of growth and development-related cis-acting elements in the *BBR-BPC* promoters of Subfamily I members might influence the growth and development of *B. napus*. Furthermore, the conserved motif analysis indicates that most members of Subfamily I are characterized by seven conserved motifs (15, 7–8), hinting at the potential roles of these motifs in plant growth and development processes. It is noteworthy that cis-acting elements such as TCT-motif, ARE, CGTCA-motif, TGACG-motif, and ABRE are widespread across different members of the BBR-BPC gene family, suggesting that these elements could play crucial roles in the fundamental life processes of *B. napus*.

Previous studies mainly focused on the potential roles of the BBR-BPC gene family during plant growth. Nevertheless, there has been less research on their expression patterns and trends under adverse conditions. Further analysis revealed that members within each subfamily exhibit different patterns of change under stress conditions, which may be due to differences in conserved motifs and cis-acting elements among subfamily members. Interestingly, under drought and hormone stress, these genes displayed certain similarities in expression patterns, which might be attributed to the plants’ shared mechanisms of adjusting osmoregulatory substances to cope with stress. Among the members of different subfamilies, *BnBBR-BPC15*, *BnBBR-BPC20*, and *BnBBR-BPC25* exhibited notably high expression levels, suggesting their significant roles in *B. napus*’s response to abiotic stresses. In addition, during the stress period, we found that the physiological state of seedlings in the early stages of salt and drought stress treatment was similar to that of untreated seedlings, and the expression level showed an upward trend. However, as the stress time increased, the expression level decreased, and the seedlings appeared wilted. This also implies that the aforementioned genes may indirectly alleviate salt ion toxicity and oxidative damage by regulating the synthesis of osmotic substances, thereby achieving a response and adaptation to adversity. However, when the treatment time increases, the plants then show irreversible damage. Notably, these three genes all originate from the C sub-genome of *B. napus*, providing insights for future functional studies of sub-genomes in this species, and can be modified by gene engineering methods, introduced into model crops such as *A. thaliana* or constructed yeast libraries, etc., to further determine whether they are potential candidate genes. At the same time, future studies could look into combining cis-acting elements, important functional motif areas (Motif 1, Motif 2, Motif 4, and Motif 8), and gene expression patterns to develop functional markers that can be used in subsequent molecular marker-assisted breeding and resistance breeding.

## 5. Conclusions

25 *BBR-BPC* genes in *B. napus* were identified and divided into three subfamilies based on their phylogenetic relationships. Each subfamily contains highly conserved Motif elements, while there are distinctive Motif elements between subfamilies; collinearity analysis highlighted that the BBR-BPC gene family underwent strong purifying selection following WGD events. In the promoters of most *BBR-BPC* genes, elements involved in plant growth processes (GT1-motif, G-box, TCT-motif), stress responses (ARE), and plant hormone responses (CGTCA-motif, TGACG-motif, ABRE) are significantly enriched. Transcriptomic results suggest that subfamilies I and V of the BBR-BPC family may play crucial roles in their growth processes. Further qRT-PCR analysis emphasized the potential significance of *BnBBR-BPC15*, *BnBBR-BPC20*, and *BnBBR-BPC25* in the response of *B. napus* to abiotic stresses. Due to the quality of the reference genome and the spatiotemporal variability of transcriptome sequencing data, this also greatly limits the excavation of functional genes. As the quality of genome sequencing continues to improve, the integration of multi-omics approaches and the application of gene-editing technologies will enable deeper insights into the BBR-BPC gene family, particularly regarding its roles in plant growth and development, responses to abiotic stress, and plant polyploidization events. 

## Figures and Tables

**Figure 1 genes-16-00036-f001:**
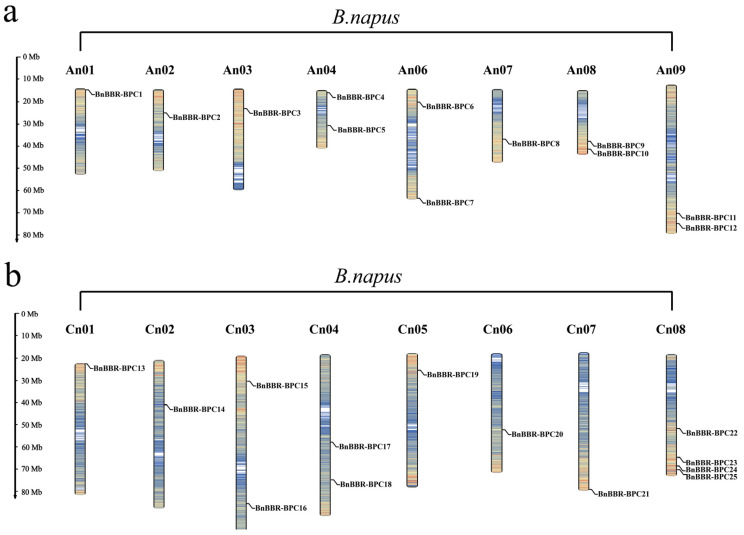
Distribution characteristics of BBR-BPC family members on chromosomes in *B. napus*. (**a**) A sub-genome of *B. napus*; and (**b**) C sub-genome of *B. napus*. Scale on the left indicates chromosome length, and color intensity of the chromosomes represents gene density.

**Figure 2 genes-16-00036-f002:**
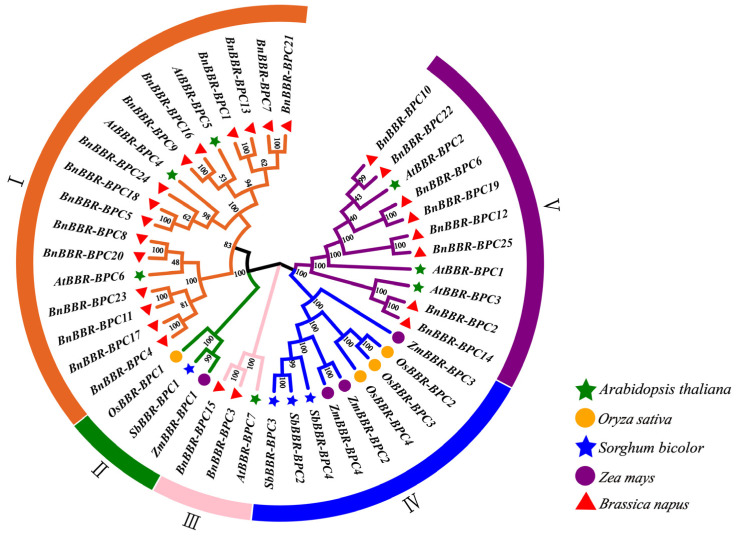
Phylogenetic analysis of the BBR-BPC gene family in *A. thaliana*, *O. sativa*, *S. bicolor*, *Z. mays*, and *B. napus*. Different shapes represent different species, while branches of different colors represent different subfamilies.

**Figure 3 genes-16-00036-f003:**
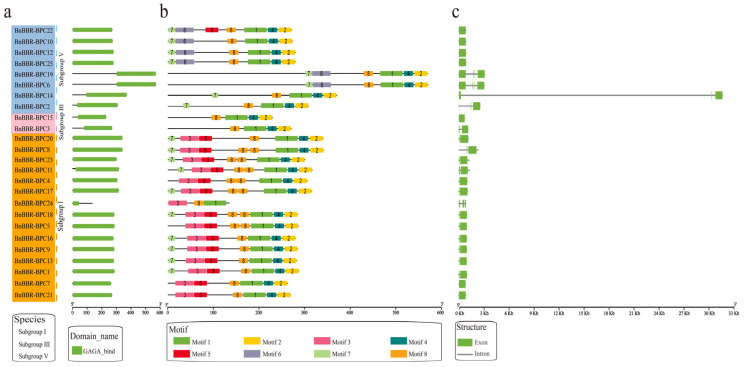
Conservation analysis of the BBR-BPC members in *B. napus*. (**a**) Conserved domains of the BBR-BPC members; (**b**) conserved motifs of the BBR-BPC members, with each motif represented by different colors and numbers; and (**c**) gene structure of the BBR-BPC members.

**Figure 4 genes-16-00036-f004:**
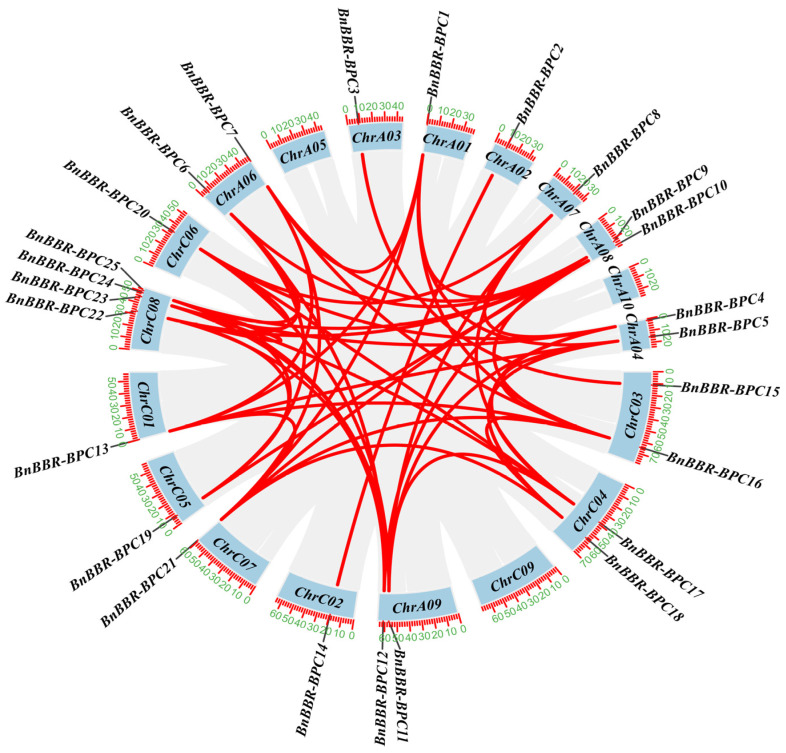
Intragenomic collinearity analysis of *B. napus*, with duplicated gene pairs within BBR-BPC gene family connected by red lines.

**Figure 5 genes-16-00036-f005:**
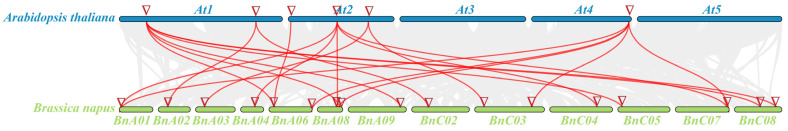
Intergenomic collinearity analysis between *B. napus* and *A. thaliana*. Collinear genes are connected by red lines, and triangles indicate BBR-BPC genes on respective chromosomes.

**Figure 6 genes-16-00036-f006:**
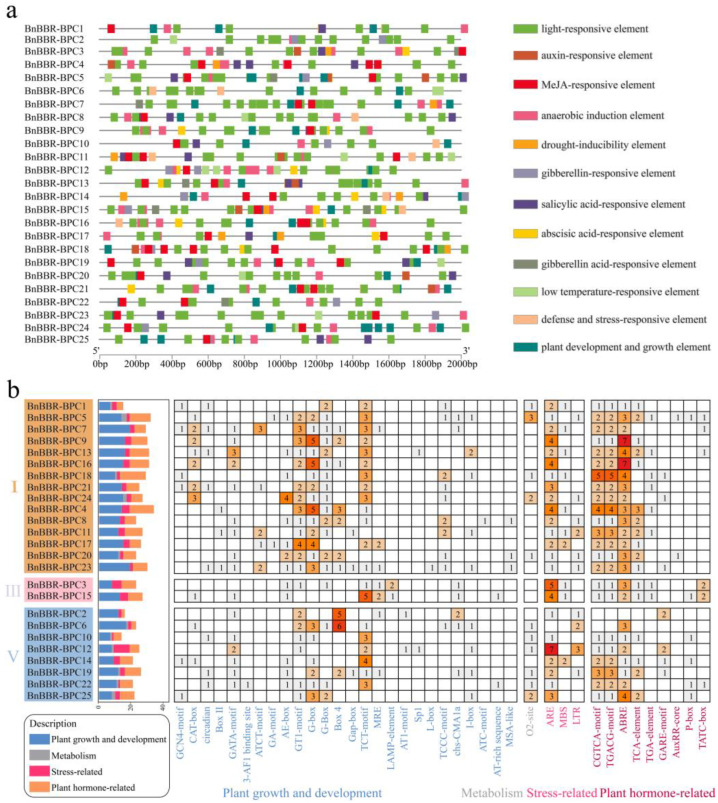
The analysis of the cis-acting element characteristics of the BBR-BPC gene family members. (**a**) The distribution and arrangement of cis-acting elements in the promoter regions, with different elements represented by colored boxes; and (**b**) a heatmap displaying the number of cis-acting elements contained by members of different subfamilies, where the bar graph shows the total content of four types of cis-acting elements, and the numbers and shades within the boxes indicate the specific quantities of each element.

**Figure 7 genes-16-00036-f007:**
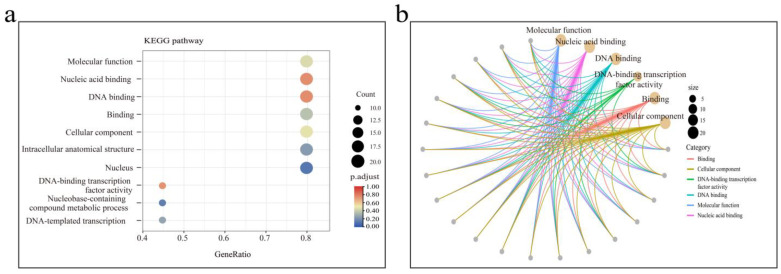
Enrichment analysis of BBR-BPC genes in *B. napus*: (**a**) enrichment in different pathways and (**b**) enrichment in key pathways and members across different subfamilies.

**Figure 8 genes-16-00036-f008:**
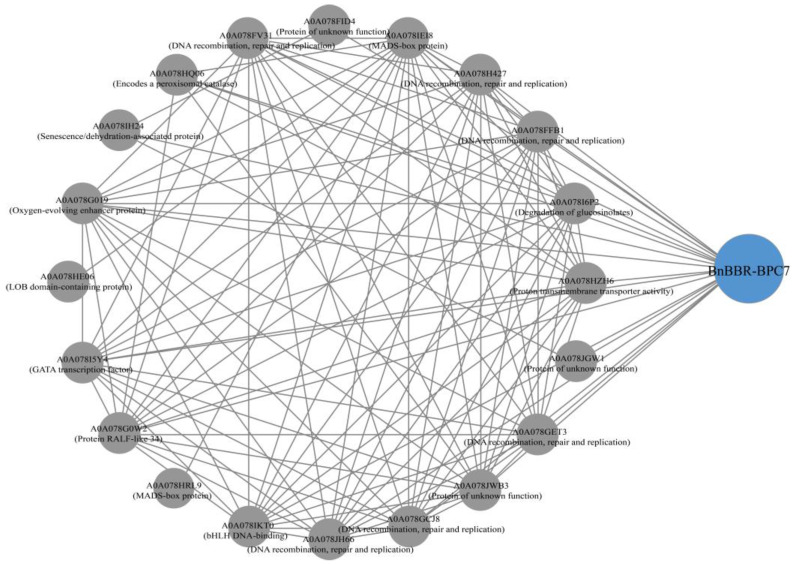
Predicted interaction network of the *B. napus* BBR-BPC gene family, where lines represent existing interactions and circles denote potential interacting proteins.

**Figure 9 genes-16-00036-f009:**
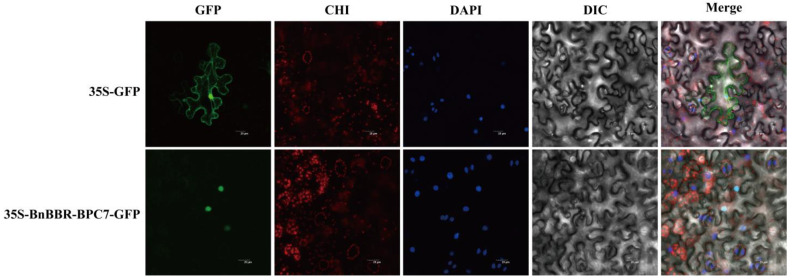
Subcellular localization analysis of *B. napus* BnBBR-BPC7 protein, with GFP, CHI, DAPI, DIC, and Merge representing green fluorescent field, chlorophyll autofluorescence, DAPI stain (a type of nuclear stain), bright field, and composite field, respectively; scale bar represents 20 μM.

**Figure 10 genes-16-00036-f010:**
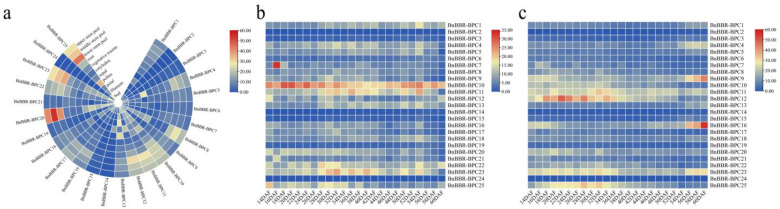
Transcriptomic analysis of the *BBR-BPC* members in *B. napus*. (**a**) Expression levels of *BnBBR-BPCs* in various tissues (bud, filament, petal, pollen, sepal, cotyledon, vegetative rosette, root, and stem peel); (**b**) expression levels in siliques at different developmental stages; and (**c**) expression levels in seeds at different developmental stages.

**Figure 11 genes-16-00036-f011:**
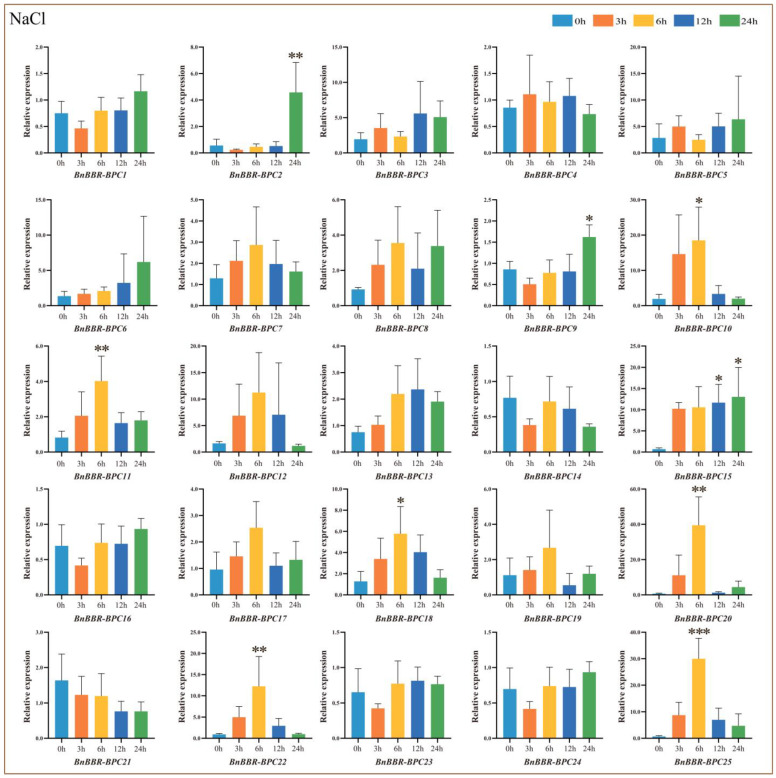
Relative expression levels of *BnBBR-BPC* members under different durations of salt stress treatment (0 h, 3 h, 6 h, 12 h, 24 h). The mean (±SE) expression values were calculated from three independent biological replicates and three technical replicates (*, *p* < 0.05; **, *p* < 0.01; ***, *p* < 0.001).

**Figure 12 genes-16-00036-f012:**
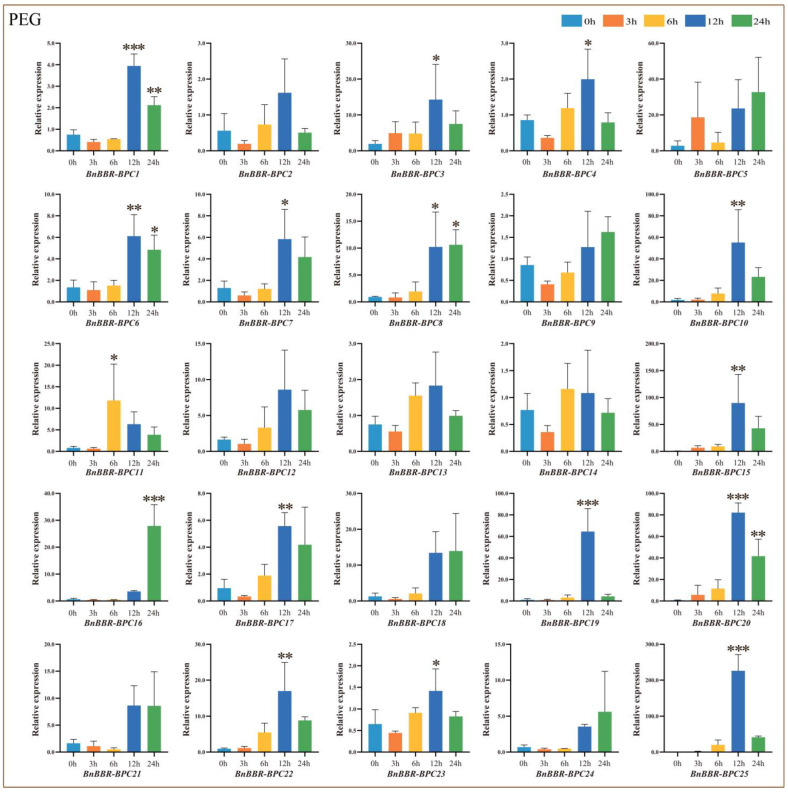
Relative expression levels of *BnBBR-BPC* members under different durations of drought stress (0 h, 3 h, 6 h, 12 h, 24 h). The mean (±SE) expression values were calculated from three independent biological replicates and three technical replicates (*, *p* < 0.05; **, *p* < 0.01; ***, *p* < 0.001).

**Figure 13 genes-16-00036-f013:**
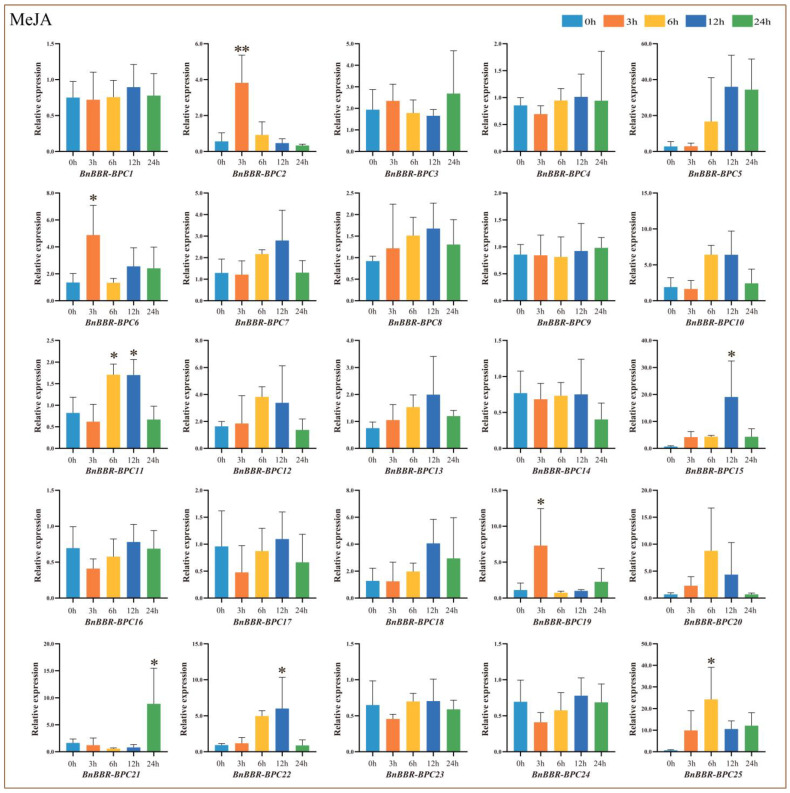
Relative expression levels of *BnBBR-BPC* members under different durations of hormone stress (0 h, 3 h, 6 h, 12 h, 24 h). The mean (±SE) expression values were calculated from three independent biological replicates and three technical replicates (*, *p* < 0.05; **, *p* < 0.01).

**Table 1 genes-16-00036-t001:** Characteristic analysis of the BBR-BPC family members in *B. napus*.

Gene Name	Gene ID	Chromosome	Number of Amino Acids (aa)	Molecular Weight (Da)	Isoelectric Point (pI)	Instability Index	GRAVY	Subcellular Localization	Orthologous Gene
*BnBBR-BPC1*	*BnaA01G0002800ZS*	A01	287	32,212.38	9.44	55.38	−0.819	nucleus	*AT4G38910*
*BnBBR-BPC2*	*BnaA02G0171100ZS*	A02	308	34,783.84	9.58	49.55	−0.605	nucleus	*AT1G68120*
*BnBBR-BPC3*	*BnaA03G0169700ZS*	A03	270	29,757.7	9.92	52.33	−0.375	nucleus	*AT2G35550*
*BnBBR-BPC4*	*BnaA04G0017300ZS*	A04	305	34,288.76	9.35	59.04	−0.922	nucleus	*AT5G42520*
*BnBBR-BPC5*	*BnaA04G0137800ZS*	A04	286	31,885.16	9.17	34.87	−0.699	nucleus	*AT2G21240*
*BnBBR-BPC6*	*BnaA06G0096700ZS*	A06	570	65,234.18	8.66	55.78	−0.791	nucleus	*AT1G14685*
*BnBBR-BPC7*	*BnaA06G0450000ZS*	A06	269	30,325.61	9.27	56.31	−0.842	nucleus	*AT4G38910*
*BnBBR-BPC8*	*BnaA07G0211100ZS*	A07	341	38,028.13	9.28	52.71	−0.859	nucleus	*AT5G42520*
*BnBBR-BPC9*	*BnaA08G0198900ZS*	A08	284	31,838.19	9.29	54.63	−0.866	nucleus	*AT4G38910*
*BnBBR-BPC10*	*BnaA08G0270900ZS*	A08	273	30,490.56	9.51	57.73	−0.840	nucleus	*AT1G14685*
*BnBBR-BPC11*	*BnaA09G0548200ZS*	A09	316	35,778.63	9.47	50.79	−0.876	nucleus	*AT5G42520*
*BnBBR-BPC12*	*BnaA09G0631800ZS*	A09	279	31,216.44	9.65	53.63	−0.853	nucleus	*AT1G14685*
*BnBBR-BPC13*	*BnaC01G0002300ZS*	C01	282	31,760.9	9.46	52.04	−0.828	nucleus	*AT4G38910*
*BnBBR-BPC14*	*BnaC02G0219700ZS*	C02	370	41,534.44	9.55	52.48	−0.561	nucleus	*AT1G68120*
*BnBBR-BPC15*	*BnaC03G0196600ZS*	C03	229	25,501.36	9.75	58.04	−0.601	nucleus	*AT2G35550*
*BnBBR-BPC16*	*BnaC03G0671500ZS*	C03	279	31,413.76	9.4	55.93	−0.861	nucleus	*AT4G38910*
*BnBBR-BPC17*	*BnaC04G0278000ZS*	C04	315	35,646.12	9.25	51.23	−0.979	nucleus	*AT5G42520*
*BnBBR-BPC18*	*BnaC04G0428300ZS*	C04	284	31,618.83	9.05	37.51	−0.708	nucleus	*AT2G21240*
*BnBBR-BPC19*	*BnaC05G0118800ZS*	C05	569	65,338.51	8.92	57.3	−0.764	nucleus	*AT1G14685*
*BnBBR-BPC20*	*BnaC06G0222600ZS*	C06	340	38,057.17	9.2	52.3	−0.869	nucleus	*AT5G42520*
*BnBBR-BPC21*	*BnaC07G0547100ZS*	C07	269	30,443.6	9.26	58.41	−0.863	nucleus	*AT4G38910*
*BnBBR-BPC22*	*BnaC08G0230300ZS*	C08	271	30,205.25	9.51	54.96	−0.845	nucleus	*AT1G14685*
*BnBBR-BPC23*	*BnaC08G0394500ZS*	C08	300	33,903.27	9.51	57.66	−1.010	nucleus	*AT5G42520*
*BnBBR-BPC24*	*BnaC08G0454400ZS*	C08	134	14,968.89	10.46	26.5	−0.753	cytoplasm	*AT2G21240*
*BnBBR-BPC25*	*BnaC08G0489500ZS*	C08	279	31,200.44	9.65	52.14	−0.837	nucleus	*AT1G14685*

## Data Availability

The *A. thaliana* genome information comes from the TAIR database (https://www.arabidopsis.org/, accessed on 1 March 2024), while the genome information for *B. napus* and its diploid ancestors is from the BRAD database (http://brassicadb.org, accessed on 1 March 2024). Transcriptome data for the BBR-BPC gene family are from the BnIR database (http://yanglab.hzau.edu.cn/BnIR, accessed on 2 March 2024). Additional data used in this study can be found in the Appendix A. All databases mentioned in this study are publicly accessible.

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
