# Peer review of "Genome-Wide Identification, Conservation, and Expression Pattern Analyses of the BBR-BPC Gene Family Under Abiotic Stress in *Brassica napus* L."

_genes, 2024, doi:10.3390/genes16010036_

Round 1

Reviewer 1 Report

Comments and Suggestions for Authors

This study investigates the BBR-BPC gene family in Brassica napus L., highlighting its structural and functional diversity, evolutionary conservation, and expression under abiotic stress. The authors identified 25 BBR-BPC genes and classified them into three subfamilies. Using bioinformatics and qRT-PCR analysis, the study demonstrates these genes' potential roles in growth, development, and stress responses, providing insights into plant polyploidization and stress tolerance mechanisms. Strengths include a comprehensive integration of genomic, transcriptomic, and experimental approaches. Thank you for the article.

-The study suggests functional redundancy in the BBR-BPC family. Do the authors believe this redundancy offers an adaptive advantage under abiotic stress conditions? How could this be experimentally validated?

-Could the qRT-PCR analysis be complemented by additional methods (e.g., proteomics or metabolomics) to confirm the functional roles of highly expressed BBR-BPC genes under stress?

-The study highlights segmental duplications as a driver of BBR-BPC family expansion. How do these duplications correlate with functional diversification? Were any novel functions identified for duplicated genes?

-How might the findings on stress-responsive BBR-BPC genes (e.g., BnBBR-BPC15, BnBBR-BPC20, BnBBR-BPC25) be applied to enhance stress resistance in breeding programs for B. napus or related crops?

-The qRT-PCR results show significant expression changes under salt and drought stress. Are these changes consistent across different B. napus cultivars, or might genetic variation affect stress responses?

-I would recommend that the discussion explicitly connect key genes' expression trends (e.g., BnBBR-BPC15, BnBBR-BPC20, BnBBR-BPC25) to their potential functional roles in abiotic stress tolerance. For instance:

  • How do the observed expression patterns translate to physiological or biochemical adaptations in B. napus?
  • Are these genes potentially involved in pathways like osmoprotection or antioxidant responses?

Also...

-Expand on how the identified stress-responsive genes (e.g., BnBBR-BPC20, BnBBR-BPC25) could inform breeding strategies:

  • Are these genes potential candidates for genetic engineering to improve stress tolerance?
  • Could marker-assisted selection be employed using the cis-regulatory motifs or gene expression profiles?

-Include a section highlighting this study's limitations (limitations of RNA-seq data, such as potential underrepresentation of low-expression genes or tissue-specific expression variability) and future research.

Author Response

Dear reviewer, we submitted it in word form, please check it.

Reviewer 2 Report

Comments and Suggestions for Authors

Dear Authors,

The manuscript titled "Genome-wide Identification, Conservation, and Expression Pattern Analyses of the BBR-BPC Gene Family Under Abiotic Stress in Brassica napus L." provides significant insights into the BBR-BPC gene family. A total of 25 genes were identified, categorized into three subfamilies, and found to share conserved motifs (Motifs 1, 2, 4, and 8). These genes are highly conserved across monocots and dicots and have undergone purifying selection following whole-genome duplication (WGD).

Promoter analysis revealed cis-acting elements associated with light, hormone responses (e.g., methyl jasmonate), and abiotic stress. RNA-seq data and functional analyses under salt, drought, and hormone stress identified BnBBR-BPC7, BnBBR-BPC15, BnBBR-BPC20, and BnBBR-BPC25 as key genes.

This study enhances understanding of the BBR-BPC gene family’s role in B. napus and its broader relevance to plant polyploidization and stress responses.

However, several critical aspects require attention to improve the manuscript's clarity, coherence, and overall impact:

Abstract

Strengths:

  • Provides a concise summary of the study's objectives, methodology, and major findings.
  • Includes key insights into evolutionary conservation and functional analyses of the BBR-BPC gene family.

Points for Improvement:

  1. Add a clearer statement of the practical implications of the findings, particularly how they advance agricultural or scientific knowledge.
  2. Simplify and clarify the description of conserved motifs and cis-acting elements for broader accessibility.

Introduction

Strengths:

  • Contextualizes the importance of the BBR-BPC gene family in plant development and stress responses.
  • Highlights the gap in knowledge regarding Brassica napus.

Points for Improvement:

  1. Reorganize content to present the broader significance of polyploid plants before focusing on the specific gene family.
  2. Simplify technical terminology for broader readership while maintaining scientific accuracy.
  3. Improve transitions between the functional roles of the BBR-BPC gene family and its evolutionary context.

Materials and Methods

Strengths:

  • Provides a detailed description of bioinformatics tools, databases, and experimental protocols.
  • Includes robust methodologies for gene identification, characterization, and expression analysis.

Points for Improvement:

  1. Clarify how threshold values (e.g., E-value < 1×10-5) were determined for homology searches.
  2. Ensure subheadings are consistent and clearly delineated for ease of navigation.
  3. Include brief justifications for each major methodological step, particularly the selection of B. napus as a model species.

Results

Strengths:

  • Comprehensive analysis of gene identification, evolutionary relationships, and functional characterization.
  • Well-structured subsections covering phylogenetic analysis, collinearity, cis-elements, and expression patterns.

Points for Improvement:

  1. Provide a more concise summary of key findings at the end of each subsection.
  2. Ensure figures are accompanied by clear, standalone legends that contextualize the findings.
  3. Highlight specific examples of gene interactions or regulatory pathways to emphasize functional significance.

Discussion

Strengths:

  • Effectively relates findings to broader biological and evolutionary contexts.
  • Discusses the potential roles of key genes under abiotic stress and hormone treatments.

Points for Improvement:

  1. Expand on the implications of functional redundancy within the BBR-BPC family.
  2. Discuss the potential applications of findings in crop breeding or stress resistance more explicitly.
  3. Minimize repetition of results and focus on integrating insights with existing literature.

Conclusions

Strengths:

  • Summarizes the study's contributions to understanding the BBR-BPC gene family.
  • Highlights the utility of combining transcriptomic and bioinformatic approaches.

Points for Improvement:

  1. Add a forward-looking statement on future research directions, including functional validation and genome editing applications.
  2. Streamline for brevity while retaining key conclusions.

References

Strengths:

  • Extensive citations from recent and relevant literature.

Points for Improvement:

  1. Ensure consistency in formatting according to journal guidelines.
  2. Remove redundant or less relevant citations to tighten the reference list.

Author Response

(The authors gave the same response as above.)
